

# New media art design based on fast visual segmentation and 3D image processing

Zhan Wang

Sanmenxia Polytechnic, Sanmenxia, China

## ABSTRACT

Acquiring innovative styles and compositions from intricate and heterogeneous artistic imagery has emerged as a pivotal research quandry within contemporary new media art image conception. In a concerted effort to adeptly distill the quintessence of artistic styles and elements embedded within these visuals, an innovative methodology is posited herein, underpinned by an enhanced U-net segmentation framework and harmoniously fused with the surface extraction image reconstruction algorithm. This meticulous amalgamation endeavors to attain accurate segmentation and tridimensional reconstruction of the artistry encapsulated in these images. Primarily, the imagery is meticulously partitioned, culminating in an output that artfully encapsulates the inherent artistic attributes. Subsequently, this segmentation outcome is adeptly reconstituted, bestowing form to a three-dimensional artistry model. Empirical validation substantiates the efficacy of this approach, with the method's Mean Intersection over the Union (MIoU) parameter yielding an impressive score of 0.939 in segmentation performance. Moreover, the peak signal-to-noise ratio and structural similarity attain commendable zeniths of 38.16 and 0.9808, respectively, underscoring the excellence of the reconstruction process. The proposed methodology demonstrates its prowess in exacting segmentation and comprehensive reconstruction of semantic intricacies and nuanced features pervading the realm of artistic imagery. Consequently, this novel methodology augments artists' capacity to discern diverse artistic paradigms and fabricate superlative new media art compositions of heightened caliber.

Corresponding author
Zhan Wang, duolala6648@163.com

## INTRODUCTION

Art images reflect people's pursuit of quality of life and spirituality and can express emotions not defined by words and language (*Zhang, Sun & Yuan, 2022*). People mainly learned and appreciated art images through art circles when the Internet was initially developed. New media technology has emerged in response to the needs of the times because of the extensive development of the Internet, significantly impacting people's lives in the economy, entertainment, politics, and other regions. In art design, new media technology has changed the way of thinking and expression for people to create art images. More and more artists have begun to use new media technology to develop and disseminate contemporary art, making art images present digital display, enriching the way for people

to understand art information, and promoting the exchange and development of art forms worldwide (*Wu, 2021*). The current research on art images focuses on the segmentation of paintings according to fixed themes and expression methods and the classification of works according to the creative style. There is less research on type and segmentation based on the style characteristics of multi-class art images (*Zhao, Jiang & Qiu, 2022*). Different painters have different ways of creating. The painting's line thickness can reveal the creators' painting style. Achieving effective and accurate classification and segmentation of multi-category art images is a problem worth studying (*Liu & Sun, 2021*).

Earlier, related researchers used traditional machine learning methods for feature extraction of art images and obtaining relevant art styles and painters' factions (*Magdalena & Robbie, 2021*). For example, *Li & Wang (2004)* utilized a two-dimensional multi-resolution Markov hybrid model to identify and compare the painter faction and segment most of the image's regions. *Sheng & Jiang (2013)* used the Sobel method to obtain art style-rich local detail features and operated an information entropy fusion algorithm to achieve painter classification. *Shen (2009)* used the radial basis function to extract overall features, shape, and local texture features. Then, they calculated the hamming distance between the features to achieve art image classification. *Sun et al. (2016)* used Monte Carlo models and vector machines to classify the features of traditional Chinese paintings. *Wang et al. (2013)* used supervised heterogeneous methods to extract textures and shapes of Chinese paintings. Still, the number of feature dimensions was too small to depict the overall features of Chinese paintings. *Jiang et al. (2006)* used edge features to distinguish between Chinese painting and calligraphy expressions and thus achieve the classification. *Gao et al. (2017)* used feature and edge detection to obtain image-critical regions, which were analyzed using a cascade classifier to derive art images. Although the above methods accomplish feature extraction and classification of art images to some extent, the features displayed in art images are often combined in organic form. The combination techniques are difficult to summarize. These traditional methods are subject to recognition errors and time-consuming problems.

Deep learning has recently been extensively applied in several fields, particularly in solving issues like hazy borders, subpar accuracy, and poor image quality (*Shao & Li, 2021*). Deep learning-based image segmentation technology has outperformed machine learning in extracting global features and local details. *Erdi et al. (1997)* proposed a thresholding algorithm for image semantic segmentation. *Li, Chen & Zhang (2019)* proposed a U-shaped network structure more suitable for fine image processing. Unlike the summation mechanism of FCN, U-Net uses multiple up-sampling and downsampling to acquire high-level semantic information gradually and stitches the dimensions of the same channels together through jump connections to achieve the fusion of features, significantly improving the performance of segmentation. U-Net has had some success with image segmentation, but it cannot extract more detailed contextual information. Therefore, more new structures with U-Net as a variant are proposed to extract and retain more essential features. For example, *Duan et al. (2018)* designed a lightweight Seg-Net model, which presents a new up-sampling method to save memory space. The U-Net++ network and variant, proposed based on U-Net, makes a more significant breakthrough in image segmentation technology

(*Zhou et al., 2018*). This network can efficiently address the issue of adaptive selection of sampling depth among different samples and accelerate feature information extraction at various levels. However, there is a problem with the abrupt increase in the number of model parameters, which raises the computational cost of the model and necessitates significant GPU resources (*Tan et al., 2021*). With the deepening of the network model, *Re, Stanczyk & Mehrkanoon (2021)* proposed an ACU-Net network-based image segmentation technique using depth-separable convolution to trim down the model's parameters. *Wang, Li & Zhuang (2021)* proposed an ATU-Net segmentation network, which adds an attention mechanism based on U-Net and uses depth-wise convolution instead of traditional convolution. *He, Fang & Plaza (2020)* proposed an updated ResU-Net model for high-resolution image segmentation. *Zhao et al. (2022)* proposed hybrid attention for effective image segmentation.

In recent years, significant advancements have been made in the field of image segmentation through the application of deep learning techniques. *Zhao et al. (2017)* introduced a novel approach using an Inceptionv3-based architecture tailored explicitly for enhancing the accuracy of segmenting small target images. This innovation addresses the challenge of accurately delineating intricate details in images with diminutive subjects. Similarly, *He et al. (2017)* proposed a scene-parsing network characterized by a pyramid structure, harnessing contextual information and leveraging global features. This pioneering design enables the comprehensive parsing of diverse scenes, achieving impressive semantic segmentation results. By integrating contextual cues, this method proves especially effective in discerning intricate scene targets, contributing significantly to the advancement of semantic segmentation technology.

Moreover, *Wan (2020)* pioneered the application of Mask R-CNN in image segmentation, achieving a remarkable fusion of high-quality semantic segmentation and object detection. This integration showcases the potential of combining these two tasks, enhancing the understanding of images and their constituent objects. The ability to simultaneously perform semantic segmentation and object detection opens new avenues for practical applications, ranging from autonomous vehicles to surveillance systems.

While these deep learning-based segmentation algorithms have demonstrated notable achievements across various images, they exhibit susceptibility to noise and external interference, which can compromise their performance. This drawback becomes particularly evident when applying these techniques to art image design. The intricacies and subjective nature of artistic elements present unique challenges, making it difficult for these algorithms to segment and reconstruct various artistic components within images accurately. Consequently, the current state of segmentation methods falls short of fulfilling the intricate demands of art image design.

In light of these limitations, there is a clear need for further research and innovation in the intersection of deep learning and art image design. Developing robust algorithms for noise, external interference and subjective artistic nuances is essential for advancing the capabilities of image segmentation techniques in creative domains. Addressing these challenges will refine the accuracy of segmentation and empower artists and designers to

harness the full potential of deep learning for creating captivating and expressive visual compositions.

To comprehensively distill the innovative artistic styles and intricate nuances inherent within art imagery, thus augmenting the artist's creative methodology and design impetus, this scholarly exposition posits an avant-garde art image segmentation algorithm anchored in an enhanced U-net framework. This pioneering algorithm engenders precise art image segmentation and subsequent reconstruction. The principal innovations encapsulated herein are outlined as follows:

Substantive enhancements are effectuated through the reasonable adjustment of U-net network parameters and the assimilation of spatial attention and parity cross-convolution mechanisms. This synthesis augments the network's segmentation efficacy and expedites its computational velocity in art image dissection.

The astute application of the isosurface extraction algorithm synergizes with the segmentation outcomes to realize the meticulous reconstruction of three-dimensional artistry, thereby facilitating the attainment of veracious 3D art images.

The structural delineation of this manuscript is as follows:

Section 'Improved Art Image Segmentation Algorithms' expounds upon the art image segmentation algorithm predicated on the refined U-net network, elucidating the novel modulations and adaptations employed to amplify its efficacy.

Section '3D Reconstruction of Artistic Image Based on Isosurface Extraction Algorithm' elucidates the methodology underpinning the three-dimensional art image reconstruction, facilitated by the equivalence surface extraction algorithm, ensuring precision in translating segmented results into tangible artistic representations.

Section 'Experiment' substantiates the algorithm's prowess through empirical demonstrations, showcasing its prowess in segmentation and subsequent reconstruction domains.

Lastly, 'Conclusion' provides a comprehensive summary, casting a retrospective gaze upon the content encapsulated within this discourse while concurrently projecting the trajectory for future endeavors.

## IMPROVED ART IMAGE SEGMENTATION ALGORITHMS

U-Net is a neural network model based on the encoder–decoder structure, which is like the shape of the letter U, and its structure is shown in Fig. 1. It has been widely used in different fields. The U-Net architecture can adapt to images of different sizes due to its fully convolutional nature. This is particularly useful when dealing with images of varying dimensions.

There are two paths in U-Net: systolic path and expansive path. The systolic path extracts the data and then the expansive path obtains the output results. The systolic path of the U-net is the same as the conventional convolutional neural network structure. ReLU and maximum pooling operations are used to double the number of feature channels.

On the expansive path, the final recovered segmented image can retain rich input information by superimposing the number of channels. Each operation step contains an

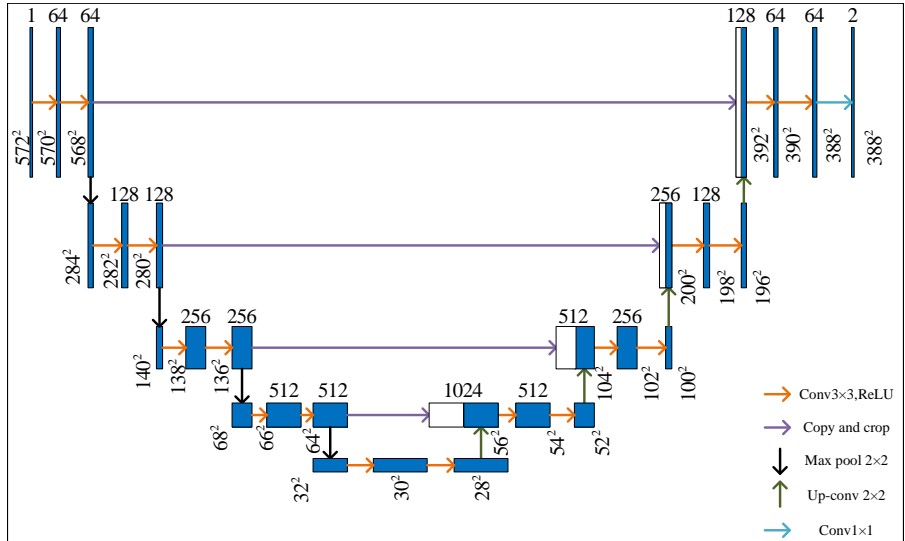

**Figure 1** U-Net network structure.

up-sampling to halve the number of channels. Then, the feature maps of the same layers in the systolic path are stitched together by copying and cropping, then using two 3 × 3 convolutions and correcting the linear units. In the last layer, the feature vectors of the elements are mapped to different classes using 1 × 1 convolution. This U-Net network has 23 convolution layers. This structure design can effectively extract image details, which is conducive to its application in semantic segmentation. However, when dealing with complex art images, U-Net is difficult to show a good segmentation effect.

## Improved U-net

Figure 2 depicts the improved U-Net network's organizational structure. Set the U-Net network input as a single channel, with a classification number of 5, to meet the requirements of identifying art images and unify all training art images to 256 × 256 resolution, adding spatial attention and parity cross convolution. Training a deep learning model on high-resolution images can be computationally intensive and time-consuming. The computational burden is reduced by resizing all training images to a standardized resolution, such as 256 × 256, enabling faster iterations during training. Spatial attention and parity cross-convolution can effectively enhance the features of art images and expand the perceptual field, thus improving the segmentation accuracy of the network. The setting of multiple convolution layers can help the network fully extract and identify features.

## Spatial attention

In visual segmentation tasks, good image features can significantly improve the segmentation accuracy of the algorithm. Convolutional neural networks have long been the mainstream selection approach for image segmentation. From the development history of convolutional neural networks, good graph features largely determine the accuracy of image segmentation tasks. The feature maps in convolutional neural networks are usually

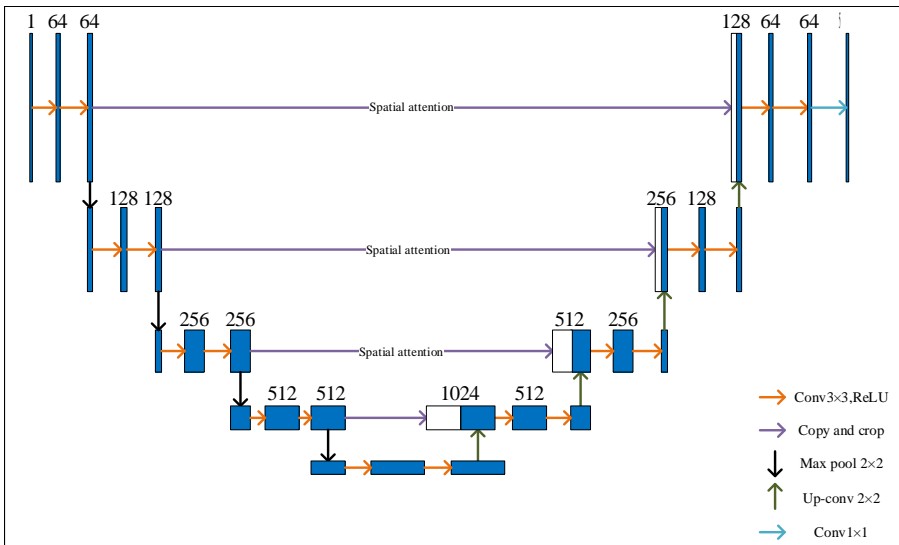

**Figure 2** **Improved U-Net network structure.**

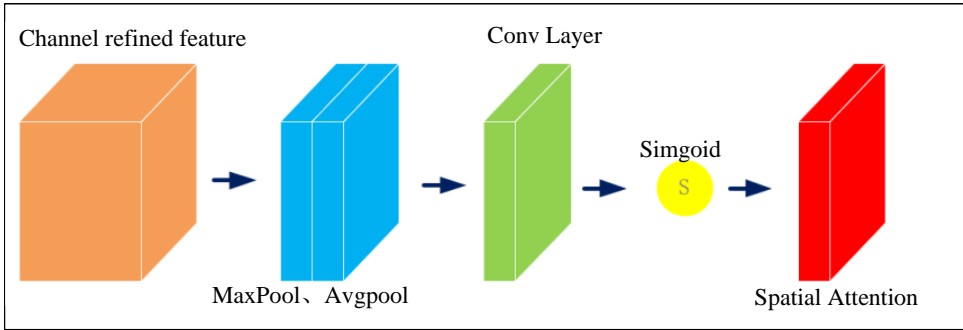

**Figure 3** **Spatial attention structure.**

segmented into different regions. Each region of the feature map has a very different contribution to the result for a particular image segmentation task. Therefore, it is necessary to increase the number of features extracted from the target region by the network and to suppress invalid or inefficient features to obtain more beneficial feature information for a particular visual task. Spatial attention16 is an optimization scheme designed to simulate human thinking, and its core idea is to redistribute resources based on contribution values, which can improve the attention of neural networks to target regions. Since the study aims to segment various elements in the art image, spatial attention is introduced to change the weight values of different feature regions of the image. The feature regions with high relevance will have higher weights so that the target regions needed for the task will be more prominent, helping the network to acquire more features and achieve better segmentation results. Figure 3 shows its structure.

The specific implementation of the spatial attention is to max pooling and average pooling of the feedback characteristic graph to get the results $X$ max and then splice the

results to get the composite signal *Xconcat*. $1 \times 1$ convolution is used to one-dimensional the signal to obtain the weight graph. The final output signal is the input signal affected by the weight chart, as shown in Formula (1) and Formula (2).

$$Xconcat = Concat\,(X\max + Xavg) \tag{1}$$

$$Xout = Xin \cdot \delta\,[conv(Xconcat)] \tag{2}$$

Formulas (1) and (2) splice the results to obtain composite signals *Xconcat*. $1 \times 1$ convolution is used to one-dimensional the signal to obtain the weight graph. The final output signal is the input signal affected by the weight chart.

## Parity cross convolution

Within image segmentation, the perspicacious delineation of objects stands as the focal point and crux of this endeavor. In art imagery, the inherent characteristics often lack the conspicuousness requisite for facile recognition, thereby conferring an inherent complexity to the segmentation process. In the paradigm of U-net architecture, as one delves deeper into the strata of network layers, the convolutional layers bequeath semantic information of substantial potency, thus furnishing a promising foundation for the precision of target segmentation. However, this advantage is juxtaposed with a miniature perceptual field that inadvertently elides myriad intricate details.

Conversely, the shallower convolutional layers yield a more expansive perceptual field, harboring a profusion of intricate target particulars that bear relevance to the task of specific segmentation. Regrettably, a paucity of semantic import undercuts this detail bounty and impairs intricate segmentation objectives. Compounding this predicament is the quandary emanating from the augmented resolution of features gleaned from deeper network strata. In contrast, it seemingly augments potential. It frequently constricts the target's mapping scope to select diminutive and deep-seated feature maps, consequently impinging upon the targeted segmentation's exactitude.

The singular integration of spatial attention into the U-net architecture, although beneficial in accentuating feature relevance, regrettably neglects the amplification of the perceptual field. Moreover, U-net's utilization of stepwise odd convolution for its downsampling stratum engenders a notably skewed distribution of the perceptual domain, thereby stifling the network's capacity to distill granular semantic information and consequently imperiling the accuracy of segmentation.

To ameliorate these complexities, this article introduces an innovative convolutional framework that synergistically melds even and odd convolutional layers, thereby engendering a parity cross-convolution schema. The efficacy of this integration is epitomized as follows:

$$Y = F3(\sigma(F2(X))) \tag{3}$$

$X$, $Y$ indicates the input and output, respectively. $F3$ and $F2$ denote the $3 \times 3$ and $2 \times 2$ convolution layers, respectively. $\sigma$ is the nonlinear operation.

To reduce the complexity further, a $2 \times 2$ even convolution layer is constructed before the U-net convolution layer in this article to realize the connection adjustment for the

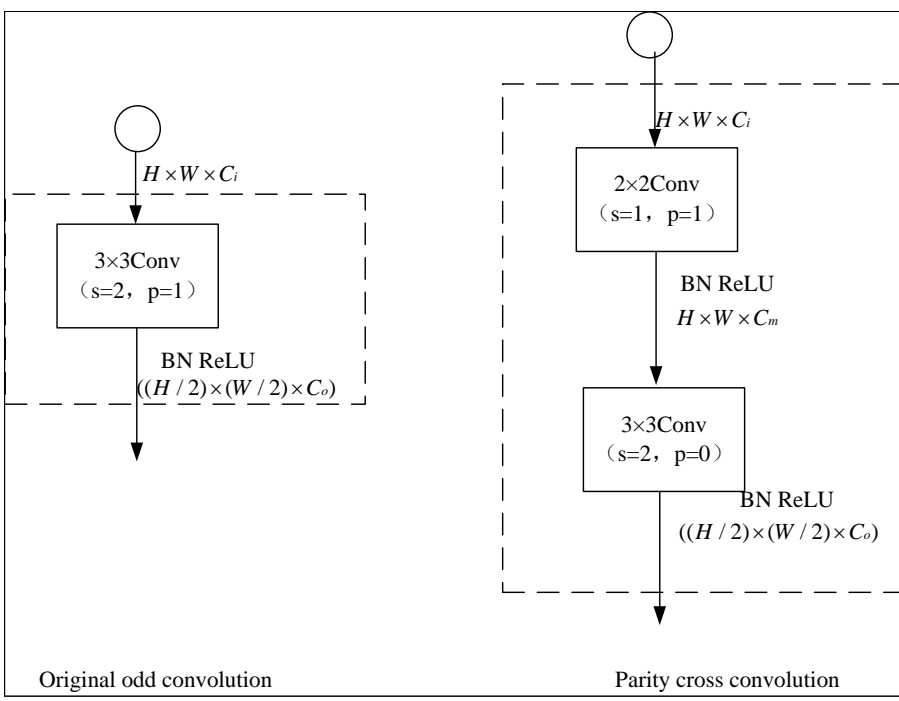

**Figure 4** **Original odd convolution and parity cross convolution.**

subsequent downsampling. Figure 4 shows the stepwise odd convolution and the parity cross convolution. The convolution of $2 \times 2$ has the lowest computational complexity compared to other convolution layers, so it can minimize the time complexity based on the effective elimination of perceptual field inhomogeneity. In the parity cross convolution, the padding parameters of the convolution are set to 1 and 0, respectively. With an even number of inputs, this padding eliminates the output asymmetry caused by the stepwise odd convolution.

The perceptual field distribution after using the parity cross-convolution is given in Fig. 5, where the input image is $6 \times 6$ and the feature map is $3 \times 3$. After using the parity cross convolution, the perceptual field intensity distribution of each point of the input tends to be homogeneous, which can effectively avoid the systematic bias caused by the stepwise odd convolution, thus making full use of each pixel and finally improving the segmentation accuracy of the image semantics.

## Loss function

The overall loss function used for the network in this article is:

$$L = \lambda 1 Lseg + \lambda 2 Ledge \tag{4}$$

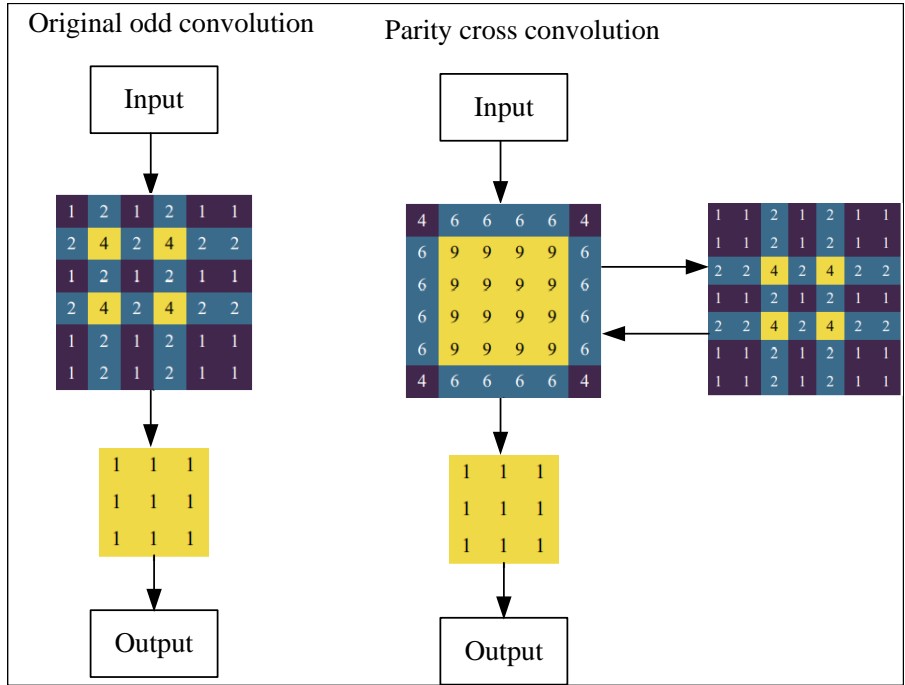

**Figure 5** Comparison of perceptual fields before and after using parity cross convolution.

where *Lseg* is the weighted loss (*Zhou et al., 2018*). The Formula is expressed as:

$$Lseg = -\sum_{c=1}^{M} wcyc \log(pc) \tag{5}$$

$$wc = \frac{N - Nc}{N} \tag{6}$$

$N$ is the total number of pixels. $Nc$ is the pixel of category $c$. $yc$ isa vector with elements taking only 1 and 0. 1 is taken if the category is the same, otherwise 0 is taken. $pc$ indicates the probability that the predicted sample belongs to $c$.

*Ledge* is the binarized cross-entropy loss function, defined as follows (*Tan et al., 2021*).

$$Ledge = -\sum_{i}(bi \log \hat{bi} + (1 - bi) \log(1 - \hat{bi})) \tag{7}$$

$bi$ and $\hat{bi}$ is the annotated image pixel values and the edge prediction feature image pixel values, respectively.

Iterative training of the network leads to an inconsistent number of results for *Lseg* and *Ledge*, so the metric scales are made consistent by weighting the coefficients. The coefficients λ1 and λ2 are set to 1 and 10, respectively.

## 3D reconstruction of artistic image based on isosurface extraction algorithm

This article reconstructs the segmented art image data in 3D images to get the 3D model of art images. Conventional reconstruction methods mainly include volume rendering

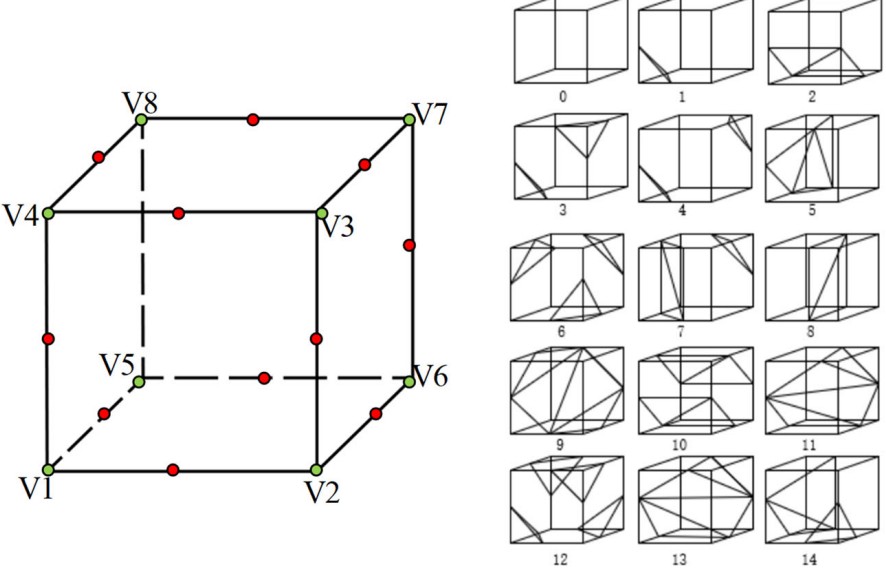

**Figure 6** The names of vertices and edges of voxel elements from isosurface extraction and the 14 topological structures.

and surface rendering. The surface rendering method has the advantages of low resource consumption and fast running speed. This article uses the classical surface rendering reconstruction algorithm-the isosurface extraction algorithm (*Zhao et al., 2022*).

As shown in Fig. 4, the central idea of the isosurface extraction algorithm is The main idea of the algorithm is to detect the voxel elements that intersect with the isosurface, calculate the coordinates of the intersection points, and then construct the corresponding grid topology for different intersections. A voxel created by eight pixels from two adjacent slices is used to locate the surface. In this article, each grid cell of the data set composed of 3D data is regarded as a voxel. For the convenience of processing, it is specified that each voxel has eight vertices. The vertex values greater than the isosurface are identified as positive; otherwise, they are. Therefore, building a table can accurately reflect 256 kinds of relationships between a voxel and the isosurface. Unifying training images to a resolution of $256 \times 256$ is a practical strategy that balances computational efficiency, memory constraints, model architecture considerations, and the trade-off between detail and context. It contributes to a stable and efficient training process, promoting generalization and optimal segmentation performance.

Since the relationship between the vertex and the isosurface is entirely symmetrical, we can reduce all cases to 128. The cube is rotationally symmetric, which can be reduced to 14 cases, and then the edge of each voxel element is indexed, as shown in Fig. 6.

The same method is used to record the relationship of edges, and then the intersection points are confirmed by linear interpolation to generate local triangles. Then, connect all the triangles to get the reconstructed 3D model. "Local triangles" refer to small triangular facets that collectively form a mesh representation of the reconstructed object's surface. Each local triangle consists of three vertices in 3D space and represents a planar segment of the

object's surface. These local triangles are crucial building blocks for constructing a complete 3D model. Isosurface images are vital in visualizing and representing the reconstructed 3D object. An isosurface image portrays a specific value (known as an iso-value) within a 3D volume dataset. In 3D reconstruction, the volume dataset is often derived from the point cloud or the voxel representation obtained during reconstruction. To create an isosurface image, a threshold value is chosen. All points or voxels with values more significant than this threshold are considered part of the object's surface. The isosurface image is then generated by connecting these surface points, effectively forming a 2D representation of the object's surface at the chosen iso-value. This image visually depicts the reconstructed object's shape, facilitating analysis and interpretation.

Specifically, there are seven steps as follows:

1. Input segmentation results.
2. Create voxels in order.
3. Calculate the index by comparing the vertex size of the voxel and isosurface;
4. Use the index to find the corresponding parameter from the table.
5. Repeat step 3 to calculate the edge.
6. Generate local triangles according to vertex and edge conditions.
7. Combining triangles and drawing isosurface images.

# EXPERIMENT

The art image dataset used in this article is obtained from the Artlib library and DaiYi (http://www.dayi.com/), among which 339 prints, 344 Chinese paintings, 340 oil paintings, 337 gouache paintings and 339 watercolor paintings are used. Accuracy, MiOU, FPS and parameter quantity are selected as evaluation indicators.

## Training model

The in-depth learning platform of this experiment is Python 1.7.0. NVIDIA GTX2080ti GPU is used for training. Five thousand photos are selected from VOC2007 to train the improved U-net network. The training curve is shown in Fig. 7. The images are converted from RGB format to grayscale images to facilitate the training. Optimized training using the RMSprop algorithm.

The RMSprop algorithm is improved based on the AdaGrad algorithm (*Li, Chen & Zhang, 2019*). The principle determines a global learning rate, and the global learning rate is continuously divided by the square root of the sum of squares of the historical gradients controlled by the decay coefficient in each learning. The RMSprop algorithm formula is as follows.

$$Sdw = \beta Sdw + (1-\beta)dw^2 \tag{8}$$

$$Sdb = \beta Sdb + (1-\beta)db^2 \tag{9}$$

$$w = w - \alpha \frac{dw}{\sqrt{Sdw + \varepsilon}} \tag{10}$$

$$b = b - \alpha \frac{db}{\sqrt{Sdb + \varepsilon}}. \tag{11}$$

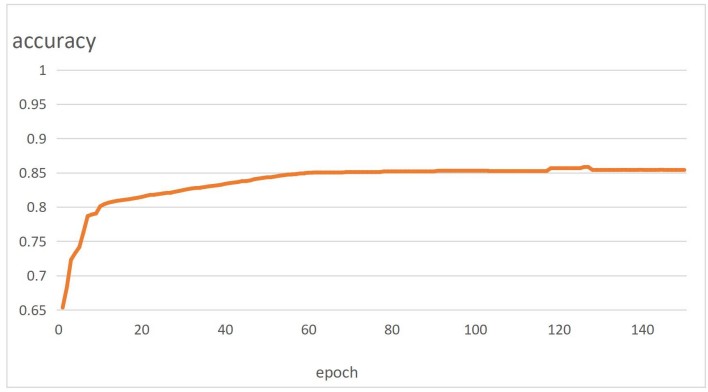

**Figure 7 Network segmentation accuracy curve.**

**Table 1 MIoU parameters comparison.**

| Network name | Artistic elements | Background | Total |
|---|---|---|---|
| U-net (*Li, Chen & Zhang, 2019*) | 0.818 | 0.995 | 0.906 |
| Dense U-net (*Zhou et al., 2018*) | 0.839 | 0.995 | 0.917 |
| ResU-net (*He, Fang & Plaza, 2020*) | 0.841 | 0.995 | 0.918 |
| Ours | 0.882 | 0.997 | 0.939 |

Among them, *Sdw* and *Sdb* represent the gradient momentum of weight *w* and offset value *b* in iteration, $\alpha$ represent the learning rate, $\beta$ represent the super parameter, and $\beta$ represent non-zero number.

After training, the art image is segmented, and then the input segmentation results are reconstructed according to the isosurface extraction algorithm to obtain the 3D model of the art image.

## Experimental results of image segmentation

One hundred images were predicted using the training model and then the dataset was segmented and compared to evaluate the training model and obtain its mean intersection over union (MIoU) parameters (*He, Fang & Plaza, 2020*). The MIoU parameters of the conventional U-net network and its variants and the improved U-net network in this article are shown in Table 1.

The larger the MIoU value, the better the segmentation of the target. It can be seen that the algorithm in this article has achieved the highest MIoU parameters. Compared with the original U-net, the segmentation effect on elements is 6.4% higher, the segmentation effect on the background is 0.2% higher, and the overall segmentation effect is 3.3% higher. This is mainly because the attention mechanism and parity cross-convolution added in this article can extract more detailed features. The quantitative analysis of MIoU parameters shows that the improved algorithm has better segmentation ability.

**Table 2   Real time performance comparison.**

| Algorithm name | Parameter quantity/M | FPS | MIoU |
|---|---|---|---|
| FCN | 13.2 | 1.43 | 0.786 |
| ACU-Net | 11.3 | 7,1 | 0.801 |
| ATU-Net | 9.2 | 9.2 | 0.811 |
| SegNet | 4.8 | 40.88 | 0.818 |
| ENet | 29.5 | 12.67 | 0.827 |
| Ours | 10.1 | 26.29 | 0.939 |

## Comparison of different algorithms

The segmentation performance of the method used in this article needs to be further validated. It is compared with FCN14, SegNet16 ACU-Net (*Re, Stanczyk & Mehrkanoon, 2021*), ATU-Net (*Wang, Li & Zhuang, 2021*) and ENet (*Zhou et al., 2018*). The dataset of self-created art images served as the testing dataset for all experiments in this article, and the software and hardware environment and parameter settings of the investigation were the same. Table 2 gives the comparison results.

Examining the findings presented in Table 2, it becomes evident that the methodology employed in this study leads to exceptional outcomes in terms of both Mean Intersection over Union (MioU) and the number of parameters utilized. Remarkably, our approach emerges as the top performer, reflecting its prowess in achieving optimal segmentation accuracy while maintaining efficiency in parameter utilization.

Considering the frames per second (FPS) metric, which serves as a gauge for real-time performance and lightweight network representation, ENet emerges as the frontrunner, delivering commendable results in this aspect. Notably, it outperforms other methods, such as FCN, which ranks at the lower end of the FPS spectrum. This juxtaposition underscores ENet's capacity for swift real-time processing.

Our algorithm substantially enhances real-time performance compared to competing models like FCN, ACU-Net (*Re, Stanczyk & Mehrkanoon, 2021*), ATU-Net (*Wang, Li & Zhuang, 2021*), and SegNet. While the segmentation speed might not match ENet's, our approach secures a noteworthy advantage in terms of MioU, exhibiting superior segmentation accuracy and fidelity. This duality of attributes showcases our method's ability to balance segmentation precision and operational speed, thereby underscoring its suitability for various practical applications.

## 3D reconstruction experiment

To objectively reflect the image reconstruction performance of the method in this article, peak signal-to-noise ratio (PSNR) and structural similarity (SSIM) are selected as the image reconstruction quality evaluation indicators (*He, Fang & Plaza, 2020*). Select ten images from the art image data set, conduct image reconstruction quality comparison experiments with conventional methods, AWAN16 and HRNet (*Tan et al., 2021*), and evaluate the image reconstruction quality with selected evaluation indicators. The average reconstruction quality data of 10 images are shown in Table 3.

**Table 3 Comparison of reconstruction methods.**

| Reconstruction method | PSNR | SSIM | Average time consumption/s |
|---|---|---|---|
| Conventional method | 20.01 | 0.3211 | / |
| AWAN | 27.44 | 0.6988 | 4.8 |
| HRNet | 29.15 | 0.7423 | 2.3 |
| Ours | 38.16 | 0.9808 | 1.2 |

The results presented in Table 3 underscore the remarkable performance of the method proposed in this article. Our approach attains the most impressive PSNR and SSIM values, standing at 38.16 and 0.9808, respectively. Remarkably, the average time required to reconstruct a single image is 1.2 s. This efficiency significantly outshines the performance of the alternative three methods under comparison. Consequently, it becomes evident that our method excels in achieving both precision and expeditiousness in the reconstruction of art images, a testament to its efficacy and practicality in the domain.

## CONCLUSION

To comprehensively extract the nuances of artistic style and intricate features embedded within art images, facilitating artists in creating novel media artworks imbued with diverse aesthetics, this article proffers an innovative approach to art image segmentation and reconstruction. Grounded in the U-net architecture and bolstered by an isosurface extraction algorithm, this method unfolds with distinct refinements. Primarily, modifications are enacted in the U-net's output classes and structural configuration, yielding heightened celerity in image segmentation. Subsequently, the infusion of spatial attention and parity cross-convolution augments the U-net's prowess, enabling the network to extract superlative details from image attributes meticulously. This intricate process culminates in reconstructing segmentation outcomes using the isosurface extraction algorithm.

Furthermore, including the RMSprop algorithm within the network's training regimen imparts a heightened acumen to its reasoning faculties, concurrently expediting the training process. Empirical investigations validate the efficacy of this approach, showcasing its ability to achieve swift and precise art image segmentation and reconstruction. Beyond this, the method elucidated herein serves as a conduit for artists to deepen their comprehension of diverse art styles and inventive methodologies, thereby elevating their wellspring of design inspiration. As our trajectory advances, we are poised to embark upon the realization of real-time, steadfast art image segmentation and reconstruction endeavors. This forthcoming phase shall be characterized by an amalgamation of avant-garde image feature extraction and processing techniques executed from an application-oriented vantage point.

### Funding

The author received no funding for this work.

### Competing Interests

The author declares that they have no competing interests.

### Author Contributions

- Zhan Wang conceived and designed the experiments, performed the experiments, analyzed the data, performed the computation work, prepared figures and/or tables, authored or reviewed drafts of the article, and approved the final draft.

### Data Availability

The code is available in the Supplementary File.

The dataset is available at Zenodo: Rhee. Bo-A, Choi, Jongwon, Hwang Jinsoo, Hyung, & Pianzola. (2021). Dataset of Instagram images related to Korean museums [Data set]. Zenodo. https://doi.org/10.5281/zenodo.5069945.

### Supplemental Information

Supplemental information for this article can be found online at http://dx.doi.org/10.7717/peerj-cs.1640#supplemental-information.

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
