# Peer review of "New media art design based on fast visual segmentation and 3D image processing"

_PeerJ Computer Science, doi:10.7717/peerj-cs.1640_

## Round 0.1 · original submission · Major Revisions

Dear authors

Your manuscript has been reviewed by the experts in the field and you will see that they have a couple of major improvement suggestions to be incorporated. Please carefully revise the article in light of their comments and resubmit.

Please also elaborate the novelty of the paper and improve the abstract.

The language of the paper also needs further improvement.

Thank you

**Language Note:** The Academic Editor has identified that the English language must be improved. PeerJ can provide language editing services - please contact us at copyediting@peerj.com for pricing (be sure to provide your manuscript number and title). Alternatively, you should make your own arrangements to improve the language quality and provide details in your response letter. – PeerJ Staff

·

Basic reporting

To acquire the artistic essence and compositional approach from the intricate and diverse art imagery represents a crucial research endeavor within the domain of image design in new media art. In this paper, we propose an improved U-net network-based segmentation algorithm, integrated with a contour surface extraction image reconstruction algorithm, to achieve precise segmentation and three-dimensional reconstruction of art images. Initially, the image is segmented to isolate solely the artistic features, followed by reconstructing the outcome to engender a three-dimensional model of the artistic image. This research endeavor not only facilitates artists in comprehending various artistic styles within art images but also fosters the creation of higher-quality new media artistic images.

Experimental design

Overall the paper is good, However, certain aspects necessitate revision as follows:

In Section 2: Improved Art Image Segmentation Algorithms, the author should utilize more headings to enhance the organization of the model's construction process.

Greater justification for the chosen method and comparative analysis with similar approaches within this research domain is essential. Expanding the list of references pertaining to other comparable studies would be valuable, and a well-structured table may augment this section.

The rationale behind introducing the "spatial attention mechanism" needs clarification. Additionally, an explanation for unifying all training images to a resolution of 256×256 is warranted.

The description of the 3D reconstruction method requires more detailed elucidation, particularly concerning "local triangles" and "isosurface images."

Validity of the findings

In Section 4.1, clarification is needed regarding the appropriate basis for selecting initial parameters during the model training process.

In the discussion section, there appears to be a paucity of comprehensive comparisons between the outcomes of the present research and those of previously related studies.

Kindly ensure that your 'conclusion' section accentuates both the scientific value added by your paper and the practical applicability of your findings, as previously indicated. I beseech you to expound upon your conclusion with greater elaboration and specificity to achieve this objective.

Additional comments

Supplementary references are required for non-original conclusive statements. For instance, (Line 40).

Considerable attention should be given to the quality of written English. Engaging a professional proofreading service before submitting the final version is highly recommended to ensure impeccable academic English.

·

Basic reporting

In this study, an enhanced machine learning method is employed to extract the salient features of artistic images, thereby enabling an accurate and comprehensive differentiation of the stylistic attributes inherent to each type of artistic representation. Moreover, your approach addresses common labeling errors, while also reducing the time and effort invested in the classification and segmentation processes.

Experimental design

To enhance the manuscript quality, I suggest the following comments and suggestions.


While establishing an art image dataset for validating the proposed algorithm is commendable, introducing the RMSprop algorithm for network reasoning and training speed enhancement is a conventional approach and not the primary contribution of this paper.

Further elucidation is warranted to explain why the U-net network, comprising 23 convolutional layers, is particularly effective in extracting image details.

The evaluation algorithm's indicators lack detailed descriptions, and it is advisable to supplement the experimental results with more datasets and evaluation metrics.

The proposed objectives, originality, and gaps addressed in the research require stronger justifications in the final part of the introduction.

More comprehensive information about the dataset, including the specific public data access address if available, should be included.

Kindly address these comments and suggestions in the revised manuscript.

Validity of the findings

Seems valid findings , I have no comments on this section

Additional comments

The author appears to have conflated the introduction with the literature review, necessitating a more coherent presentation of the latter. The current form is deemed unacceptable.
To enhance clarity, additional explanations are required for Formula (1) and Formula (2). Introducing new references could aid readers' comprehension.

---

## Round 0.2 · Minor Revisions

Based on the input received from the technical experts, I am happy to let you know that your manuscript has been judged scientifically suitable for publication and will be formally accepted for publication once it meets the following minor corrections
The language of the manuscript needs improvement for example
at line 23, Mean Intersection over Union should be Mean Intersection over the Union.
at line 25 zeniths of 38.16 and 0.9808 respectively should be zeniths of 38.16 and 0.9808, respectively
Therefore, please carefully revise the language of the paper thoroughly and re-submit.
thank you

**Language Note:** The Academic Editor has identified that the English language must be improved. PeerJ can provide language editing services - please contact us at copyediting@peerj.com for pricing (be sure to provide your manuscript number and title). Alternatively, you should make your own arrangements to improve the language quality and provide details in your response letter. – PeerJ Staff

·

Basic reporting

Comments has been addressed

Experimental design

Comments has been addressed

Validity of the findings

Comments has been addressed

Additional comments

Comments has been addressed

·

Basic reporting

ok

Experimental design

ok

Validity of the findings

ok

Additional comments

The changes have been done by author as per reviewer's comments.

---

## Round 0.3 · accepted · Accept

Thank you for your contribution. We are pleased to recommend it for acceptance.